# DYNAMICS OF LEARNING: GENERATIVE SCHEDULES FROM LATENT ODES

## ABSTRACT

The learning rate schedule is one of the most impactful aspects of neural network optimization, yet most schedules either follow simple parametric functions or react only to short-term training signals. None of them are supported by a comprehensive temporal view of how well neural networks actually train. We present a new learning rate scheduler that models the training performance of neural networks as a dynamical system. It leverages training runs from a hyperparameter search to learn a latent representation of the training process. Given current training metrics, it predicts the future learning rate schedule with the best long-term validation performance. Our scheduler generalizes beyond previously observed training dynamics and creates specialized schedules that deviate noticeably from common parametric functions. It achieves SOTA results for image classification with CNN and ResNet models as well as for next-token prediction with a transformer model. The trained models are located in flatter regions of the loss landscape and thus provide better generalization than those trained with other schedules. Our method is computationally efficient, optimizer-agnostic, and can easily be layered on top of ML experiment-tracking platforms. An implementation of our scheduler will be made available after acceptance.

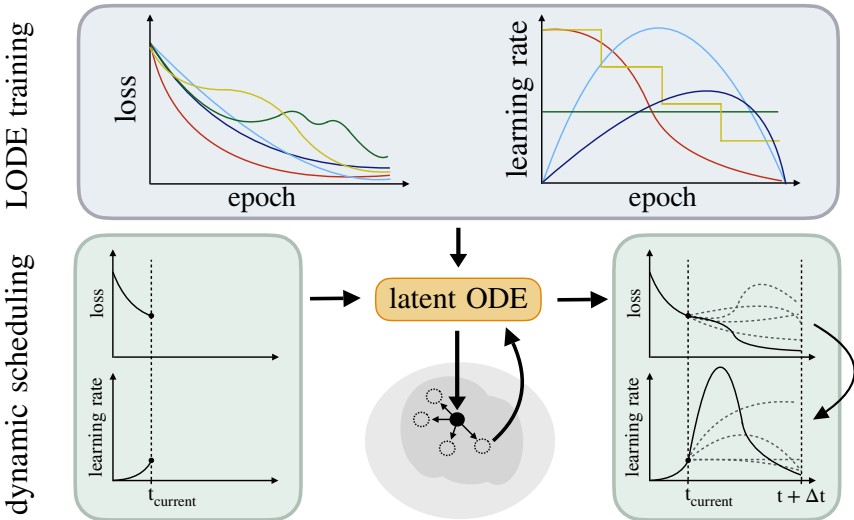

Figure 1: Schematic of the dynamic learning rate scheduler. **Latent ODE model:** Metrics tracked during a hyperparameter sweep contain a variety of learning rate schedules as well as resulting training and validation losses. We train a latent ODE model to reconstruct these training time series. **Dynamic scheduling:** For a new training run, we encode the current loss and learning rate time series with the latent ODE model, sample an ensemble in latent space centered on the current position, generate their future training behavior, and select the learning rate schedule with the best final validation performance.

# 1 INTRODUCTION

Deep learning owes much of its success to the surprising effectiveness of gradient descent for high-dimensional, non-convex optimization. The simple update rule

$$\boldsymbol{\theta}_{t+1} = \boldsymbol{\theta}_t - \eta_t \mathbf{g}_t \tag{1}$$

for the network parameters $\boldsymbol{\theta}$ and the gradient $\mathbf{g}$ (possibly after averaging, clipping, or scaling) is easy to compute, but immediately begs the question: how to set the learning rate $\eta$ and its possible variation with time $t$, the schedule.

In second-order optimization, one can set $\eta$ proportional to the inverse Hessian of the loss function, but that is computationally unfeasible for deep neural networks. And while quasi-Newton methods can approximate the inverse Hessian under certain conditions, in deep learning one usually adopts first-order methods with simple constant, oscillating, or decaying functions as schedule. The parameters of the schedule are determined by a hyperparameter search (often on a parameter grid), where the target architecture is trained repeatedly to determine a supposedly optimal $\eta_t^*$ schedule, which is then adopted for training of the production model or a model ensemble (Wu et al., 2018).

But a fundamental understanding of what constitutes the optimal learning rate schedule for a specific model–data–task combination remains elusive (e.g. Gotmare et al., 2018; Li & Arora, 2019; Defazio et al., 2023). It is therefore likely that neural networks could be trained more effectively and perform better if we understood what a good choice of $\eta_t$ needs to achieve. Several directions have been proposed to improve the optimization procedure: Hypergradient descent (Baydin et al., 2017) updates a base learning rate using its own gradient signal. Reinforcement learning (RL) methods treat the scheduler as an agent, learning a policy on the basis of the current network state and the outcomes of prior training runs (Xu et al., 2017; 2019; Wen et al., 2020; Xiong et al., 2022). Schedule-free optimization (Defazio et al., 2024) uses momentum averaging at each training step to remove the need for an explicit stopping time for the training schedule.

However, none of these techniques have a comprehensive view of the temporal evolution of the training process. We present the first scheduling technique that expresses the training performance of neural networks as a dynamical system. It leverages training runs from a hyperparameter search to learn a latent representation of the training process as a Latent Ordinary Differential Equation (LODE; Chen et al., 2018; Rubanova et al., 2019), and then predicts the learning rate schedule $\eta(t)$ with the best long-term validation performance. Our scheduler adapts to different network initializations and consistently outperforms any schedule from the hyperparameter search as well as more advanced gradient-dependent or RL schedules. It is computationally efficient, optimizer-agnostic, and can make use of experiment metrics tracked on platforms like Weights & Biases or MLFlow (Zaharia et al., 2018; Biewald, 2020).

Our main contributions are:

- We introduce a novel adaptive learning rate scheduler based on a latent ODE model to predict future training loss, validation performance, and learning rate trajectories.

- Our scheduler generates specialized schedules beyond observed training dynamics, which achieve superior test accuracy for CNN and ResNet models on the Fashion-MNIST, CIFAR-100, and ImageNet datasets and superior next-word prediction accuracy with a transformer model.

- Our method provides a practical and efficient way of navigating the multi-scale, fractal loss landscapes of neural networks (Ma et al., 2022; Ly & Gong, 2025) by pushing learning rates beyond the "edge of stability" (Cohen et al., 2021; Arora et al., 2022) and then settling in flatter minima than other schedules.

# 2 RELATED WORK

**Parametric LR schedules.** The simplest explicit schedule adopts a constant learning rate, but has been shown to produce sub-optimal results (Smith et al., 2017; Gotmare et al., 2018; Wu et al., 2019). Consequently, time-varying parametric learning rate schedules have been employed for decades (Darken & Moody, 1990; Darken et al., 1992). Among the most successful of these schedules

are cosine-decay, OneCycle (Smith & Topin, 2017, often as cosine-OneCycle) and step-decaying schedules. Often these schedules are combined with other procedures such as warm-up to achieve their best results (Loshchilov & Hutter, 2016). A drawback of these methods is that they do not not adapt to the training performance, instead they choose the schedule shape and the corresponding hyperparameters either ad-hoc or through an extensive hyperparameter optimization.

**Meta-optimizer.** Baydin et al. (2017) propose hypergradient descent, a method for dynamically adapting the learning rate during training by computing the gradient of the loss with respect to the learning rate itself. This method works for any general gradient-based optimization problems, and operates in an online, memory-efficient manner using the already computed values from the reverse-mode automatic differentiation. The method improves convergence speed and robustness across a range of tasks, particularly early in training, but remains unaware of the possible future performance.

**Reinforcement learning.** Xu et al. (2019) and Xiong et al. (2022) introduce similar RL-based approaches for adaptive learning rate scheduling, where a controller network is trained to adjust the learning rate based on features extracted from the training process, such as gradient statistics and validation loss. Rather than predicting the learning rate directly, the controller outputs a scaling factor applied to a pre-computed base schedule. While the method shows improved performance over fixed schedules on datasets like Fashion-MNIST and CIFAR-10, it requires information from the parameters of the trained network to determine the current state, a potential problem when scaling to large networks. In addition, because rewards are based on changes of the loss and loss curves become increasingly shallow, late-time actions are only weakly constrained by training runs, so the method becomes unstable unless the adjustments of the learning rate are restricted to the narrow interval $[0.9, 1.1]$, foregoing more decisive changes to the learning rate even if those would be effective.

**Schedule-free optimization.** Defazio et al. (2024) introduce an optimization framework that eliminates the need for predefined learning rate schedules and stopping time $T$. This is achieved through a novel form of momentum averaging. Their approach achieves strong empirical performance across vision, language, and recommendation tasks without requiring knowledge of the total training horizon or manual tuning of schedules, performing on-par or better than the baseline cosine and exponential-decay schedules. However, it still requires hyperparameter tuning and, because it does not depend on $T$, achieves almost all of its performance gains early during training.

**Learning curve extrapolation.** Ding et al. (2025) investigate the possibility of predicting late-stage performance of the optimization by extrapolating earlier training and test performance of MLP and CNN models. This is achieved through an initial encoding of the training trajectories via a graph-encoder, followed by a modeling with a neural ODE to predict the dynamics. The authors use this method to rank hyperparameter choices while only observing short initial optimization run. We demonstrate similar capability in Section A.3 and extend the approach to generate new and improved schedules that have not been seen during prior optimization runs.

**Sharp/dynamic LR transitions.** Subramanian et al. (2024) study the effects of sharply changing precomputed learning rate schedules during the training of large transformer models on NLP tasks. They empirically find that such drastic changes can improve convergence and suggest there may be ways to leverage this behavior to produce more effective learning rate schedules. Our work will demonstrate an effective approach to introduce sharp changes for superior long-term results.

## 3 METHODS

We present a novel scheduler that dynamically adjusts the current learning rate based on predictions of the long-term performance under the chosen schedule. In what follows, we distinguish between the *test model*, which we seek to optimize, and the *LODE model*, which is used to learn the training behavior of the test model and choose the most effective schedule.

### 3.1 LODE MODEL

We train a LODE encoder-decoder architecture (Chen et al., 2018; Rubanova et al., 2019) to reconstruct the training behavior of the test model with metrics that are commonly logged during experiments like a hyperparameter search. In particular, the LODE model operates on three-dimensional time series $\mathbf{x}(t) \equiv (\ell(t), \nu(t), \eta(t))$, composed of the training loss $\ell$, a validation performance metric $\nu$, and the learning rate $\eta$ used by the schedule. Any suitable measure of performance computed

on the validation dataset is acceptable as $\nu$; for image classification we will choose validation accuracy in our tests below, for NLP next-token prediction accuracy. Note also that, unlike Xu et al. (2019); Xiong et al. (2022), we do not include any information about the parameters of the test model. The training trajectory is thus a representation of the performance of the test network, not of its internal state. This choice allows our approach to be scaled to neural networks of arbitrary size without incurring extra costs during training or inference.

The LODE models first encodes the training time series of the test network into the latent vector

$$\mathbf{z}_0 = \text{Encoder}_\phi \left( \{\mathbf{x}(t_j)\}_{j \in \{i-\mu,\dots,i\}} \right), \tag{2}$$

where $t_i$ is the current step of the optimizer and $\mu$ the number of previous steps considered by the encoder. We choose a RNN architecture as an encoder similar to (Rubanova et al., 2019). The latent vector $\mathbf{z}_0$ represents the joint training state, characterized by the network performance *and* the learning rate used to achieve it.

The latent vectors is advanced to any time $t > t_i$ by integrating an ODE, represented by the neural network $f_\theta(\mathbf{z}, t)$:

$$\mathbf{z}(t) = \mathbf{z}_0 + \int_{t_i}^{t} f_\theta(\mathbf{z}(\tau), \tau) \, d\tau \tag{3}$$

During LODE training, we include an $\ell_2$ penalty in the loss function to minimize the path length of the latent trajectory, which encourages a time-invariant representation $\mathbf{z}$ (Auzina et al., 2024; Sampson & Melchior, 2025). Its purpose is to ensure that states with similar training loss, learning rate, and validation metric are encoded to similar regions of latent space regardless of the time at which these values are observed.

From this latent state, one can generate the entire expected training behavior:

$$\hat{\mathbf{x}}(t) = (\hat{\ell}(t), \hat{\nu}(t), \hat{\eta}(t)) = \text{Decoder}_\psi(\mathbf{z}(t)) \tag{4}$$

If the LODE model is accurate, we know how well the test model will perform when updated with a given optimizer and schedule at any point $t$ during its training.

## 3.2 DYNAMIC LEARNING RATE SCHEDULER

We can now exploit our knowledge of the training process of the test network by changing the learning rate and predicting the resulting validation performance. Our dynamic scheduler involves five steps to generate a learning rate at time $t_*$ during the optimization of the test network, which creates the test data $\mathbf{x}_* = (\ell_*, \nu_*, \eta_*)$.

**1. Encoding** The time series segment $\mathbf{x}_*$ of length $\mu$ ending at $t_*$ is summarized by the LODE encoder into the latent vector $\mathbf{z}_0$ (Equation 2). For $t_* < \mu$, we initialize $\mathbf{z}_0$ with the data from the best-performing experiment in the LODE training data. After every $\mu$ steps, the process is repeated to yield a new $\mathbf{z}_0$. Operating on such non-interleaving segments creates latents for optimization periods without our intervention, which more closely resemble the LODE training data.

**2. Perturbation of the latent vector** We create an ensemble of $n$ latent vectors $\mathbf{z}_{0,i} = \mathbf{z}_0 + \boldsymbol{\epsilon}_i$, where $\boldsymbol{\epsilon}_i \sim \mathcal{N}(0, \sigma^2)$. Each of these latents are then integrated via Equation 3 and decoded via Equation 4, resulting in $n$ sample trajectories $\hat{\mathbf{x}}_i(t)$.

**3. Sample acceptance** Of the potential trajectories, we select those samples $i$ that satisfy the following similarity condition: $\exists k \in [0, T] : |\hat{\ell}_i(t_k) - \ell_*(t_*)| < 2 \, \text{std} \, [\ell_*(t_j)]_{j \in \{*-\mu,\dots,*\}}$, i.e. there must be at least one time $t_k$ during the optimization interval, for which the predicted loss is within 2 standard deviations of the recent losses of the test network. The requirement ensures that the selected sample is not just similar in latent space, but also close in loss to the test network at the current time. It permits that the test network reaches that loss value at a different time than the prediction and accounts for the stochasticity of the training losses.

**4. Goal conditioning** Every accepted sample $i$ has at least one timestep $t_{k,i}$ with losses comparable to the current test network. We now compute the future validation metric $\hat{\nu}_{i,*} \equiv \hat{\nu}_i(t_{k,i} + \Delta t)$ we expect to achieve by following the future learning rate schedule $\hat{\eta}_i(t)$ from $t_{k,i}$ to $t_{k,i} + \Delta t$. The

---

**Algorithm 1** Dynamic Learning Rate Scheduler

---

1: **Requires** Current time step $t_*$, final time step $T$, trained LODE model $\mathcal{M}$,
2: history of training loss, validation metric, learning rate $\mathbf{X}_* = \{(\ell_*, \nu_*, \eta_*)(t_j)\}_{t_j=t_*-\mu}^{t_*}$;
3: Configuration parameters: encoding length $\mu$, noise scale $\sigma$, ensemble size $n$, horizon $\Delta t$
4:
5: **function** LODE-SCHEDULER($\mathbf{X}_*, t_*, T, \mathcal{M}$)
6:      $\mathbf{z}_0 \leftarrow \mathcal{M}.\text{encode}(\mathbf{X}_*)$                         ▷ 1. Encode test data
7:      **for** $i = 1$ **to** $n$ **do**                         ▷ 2. Create latent ensemble
8:          **if** $i = 1$ **then** $\boldsymbol{\epsilon}_i = \mathbf{0}$ **else** $\boldsymbol{\epsilon}_i \sim \mathcal{N}(\mathbf{0}, \sigma^2)$      ▷ Keep direct latents in ensemble
9:          $\mathbf{z}_{0,i} \leftarrow \mathbf{z}_0 + \boldsymbol{\epsilon}_i$
10:          $\hat{\mathbf{X}}_i \leftarrow \mathcal{M}.\text{decode}(\mathbf{z}_{0,i};\ t = 0, \ldots, T + \Delta t)$
11:          **if** $\exists k \in [0, T] : |\hat{\mathbf{X}}_i.\ell(t_k) - \mathbf{X}_*.\ell(t_*)| < 2\,\text{std}[\mathbf{X}_*.\ell]$ **then**      ▷ 3. Select similar loss
12:              $\hat{\nu}_{i,*} \leftarrow \hat{\mathbf{X}}_i.\nu(t_{k,i} + \Delta t)$      ▷ 4. Estimate future performance
13:          **end if**
14:      **end for**
15:      $I_* \leftarrow \text{argsort}[\{\hat{\nu}_{i,*}\}]_{(1,2,3)}$      ▷ Select 3 best performers
16:      $\tilde{\boldsymbol{\eta}} \leftarrow \text{mean}_{i \in I_*}[\hat{\mathbf{X}}_i.\eta(t_{k,i} + j)]$ **for** $j = 0, \ldots, \mu - 1$      ▷ 5. Average over best samples
17:      **return** $\tilde{\boldsymbol{\eta}}$      ▷ Learning rate for the next $\mu$ steps
18: **end function**

---

forward-looking horizon $\Delta t$ is a configuration parameter that controls how greedy the scheduling algorithm will be, from maximally greedy ($\Delta t = 1$) to the longest period remaining in the optimization ($\Delta t = T - t_*$). If a sample $i$ has multiple times $t_k$ with similar losses to the test network, we choose the instance with the highest learning rate.

**5. Ensemble predictor** The final learning rate is computed for the next $\mu$ steps by averaging the three best-performing sample trajectories: $\tilde{\eta}(t_* + j) = \text{mean}_{I_*}[\hat{\eta}_i(t_{k,i} + j)]$ for $j = 0, \ldots, \mu - 1$, where $I_* = \text{argsort}[\{\hat{\nu}_{i,*}\}]_{(0,1,2)}$. The combination of trajectories improves the robustness of the predictor, especially in early phases of the optimization, where validation performance is often volatile. Because of the rejection step 3, the number of similarly performing samples may be reduced to less than three, but those remaining will have the most appropriate learning rate predictions. If no comparable sample exists, ideally because this scheduler outperforms any training run from the hyperparameter search, the current schedule will be extended for another $\mu$ steps.

We show pseudocode for the LODE-based scheduler in Algorithm 1. We also perform an ablation study for the configuration parameters $(\mu, n, \sigma, \Delta t)$ of our scheduler in Section 5.4.

## 4 EXPERIMENTAL SETUP

We compare our schedule to four non-adaptive learning rate schedules (constant, cosine-OneCycle, cosine-decay, exponential-decay), hypergradient descent (Baydin et al., 2017), the RL-based method from Xu et al. (2019); Xiong et al. (2022), and the schedule-free optimization from Defazio et al. (2024). We train a convolutional neural network (CNN), and ResNet18 (He et al., 2016) on the classification tasks of the Fashion-MNIST and CIFAR-100 datasets. We also train a larger ResNet34 (He et al., 2016) on the ILSVRC 2012 Imagenet dataset (Russakovsky et al., 2015), as well as a small (28M parameters) transformer model (Radford & Narasimhan, 2018) tested on next-token prediction in a corpus of short stories (Eldan & Li, 2023). We perform a hyperparameter sweep over a range of parameters for all LR schedulers, which is detailed in Section A.2. For the Fashion-MNIST and CIFAR100 dataset-models we use the AdamW optimizer (Loshchilov & Hutter, 2017), for the ImageNet dataset we use stochastic gradient descent, and for the transformer model we use Adam (Kingma & Ba, 2014). Once the best hyperparameters are found, we repeat training for multiple random initializations; all reported results are averaged over these initializations.

We train a latent ODE model to reconstruct the training loss, learning rate, and validation performance metric from a hyperparameter search comprising the four non-adaptive learning rate schedules. We omit schedules from hypergradient descent, RL-controller, and schedule-free methods in the training data to stay within the scope of conventional hyperparameter searches. The specific architecture and training details of the latent ODE model are described further in Section A.3.

# 5 RESULTS

## 5.1 LODE MODELLING

Figure 2 shows an example of the true (black) and reconstructed (orange) time series of training loss, learning rate, and validation accuracy for a single training run of the CIFAR-100 trial. The left columns shows the results when observing only the first 5% of the training periods (and predicting the remaining 95%), while the right columns shows the results of observing the first 50%. In both cases, we see excellent reconstruction fidelity, which demonstrates that the training behavior, i.e. the relation between and time evolution of training loss, validation accuracy and learning rate, is recognizable even from very short segments of the trial run. We show further performance analysis of the LODE model in Section A.3.

## 5.2 OPTIMIZER PERFORMANCE

We compare the final accuracy of the trained test model for the baseline schedules and our LODE scheduler in Table 1. All results represent the best hyperparameter configuration of each schedule, with results averaged over 20 random seeds for Fashion-MNIST and CIFAR100 datasets, and 10 for ImageNet and TinyStories (more details in Section A.2). We also show the probability distribution of final accuracies for these tests in Figure 3. Our LODE scheduler outperforms every baselines for every model architecture and dataset. It achieves the highest-performing results very consistently, with small spread in the test accuracy.

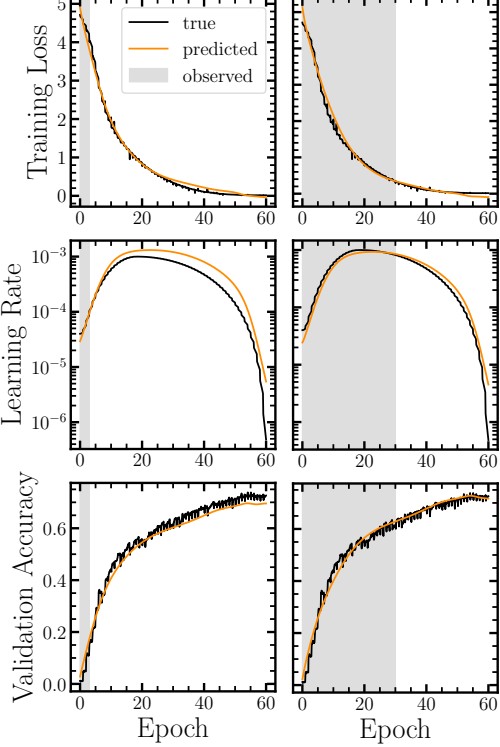

Figure 2: Training loss, learning rate, and validation accuracy observed for the CIFAR-100 trial (black) and prediction by the LODE model (orange) based on the first 5% (left) and 50% (right) of training epochs. Grey shading indicates the portion of the training trajectories encoded by the LODE model.

## 5.3 CHARACTERISTICS OF THE TRAINING PATH

In Figure 4 we plot the learning rate sequences generated by the LODE scheduler for each model/dataset combination. It shows several noteworthy features:

- The schedule shapes deviate noticeably from any parametric form in the training data.
- The change of either the model architecture on the same training data (CNN vs ResNet18 on Fashion MNIST) or of the training data for the same architecture (ResNet18 on Fashion MNIST vs CIFAR-100) leads to significant changes for high-performing schedules.
- Individual schedules show consistent patterns within each test case despite random initializations and the adaptability of the LODE scheduler.

To further explore the consequences of these new schedules, we compute the sharpness of the loss surface at convergence for the ResNet18 model on the CIFAR-100 dataset. Achieving wider, flatter minima results has been found to indicate more generalizable and better performing models (Hochreiter & Schmidhuber, 1997; Keskar et al., 2016; Jastrzkebski et al., 2017; Chaudhari et al., 2019; Petzka et al., 2021). Algorithms, such as stochastic weight averaging and sharpness-aware minimization (Izmailov et al., 2018; Foret et al., 2020) leverage this finding, even though a flatness condition alone should not be used as success metric (Dinh et al., 2017; Kaur et al., 2023).

Table 1: Comparison of final test accuracy for all schedules, models, and datasets (details in Section 4). We report the mean $\pm$ standard deviation for 20 random seeds (10 for ImageNet). For the NLP task the test accuracy is next-word prediction accuracy. Our LODE-scheduler achieves superior results in every case. *Note: RL-controller was not evaluated on ImageNet or the NLP task due to high computational cost. The ImageNet values listed here are taken from Xiong et al. (2022).

| | Fashion MNIST | | CIFAR-100 | ImageNet | NLP |
|---|---|---|---|---|---|
| | CNN | ResNet18 | ResNet18 | ResNet34 | Transformer |
| Scheduler | Test Acc (%) ↑ | Test Acc (%) ↑ | Test Acc (%) ↑ | Test Acc (%) ↑ | Test Acc (%) ↑ |
| Constant | $91.5^{\pm 0.1}$ | $91.9^{\pm 0.5}$ | $67.3^{\pm 0.6}$ | $61.7^{\pm 0.4}$ | $57.9^{\pm 0.1}$ |
| Cosine | $92.5^{\pm 0.1}$ | $93.5^{\pm 0.1}$ | $71.2^{\pm 1.4}$ | $73.9^{\pm 0.4}$ | $58.5^{\pm 0.1}$ |
| OneCycle | $92.9^{\pm 0.2}$ | $93.6^{\pm 0.1}$ | $74.0^{\pm 1.2}$ | $73.8^{\pm 0.5}$ | $58.1^{\pm 0.2}$ |
| Exp. Decay | $92.4^{\pm 0.2}$ | $93.5^{\pm 0.1}$ | $69.6^{\pm 0.9}$ | $66.1^{\pm 0.5}$ | $58.3^{\pm 0.1}$ |
| Hypergrad | $92.8^{\pm 0.1}$ | $91.7^{\pm 0.5}$ | $70.3^{\pm 1.0}$ | $59.9^{\pm 0.5}$ | $55.3^{\pm 0.4}$ |
| RL-controller | $93.0^{\pm 0.1}$ | $92.9^{\pm 0.4}$ | $73.9^{\pm 0.9}$ | $*71.6^{\pm 1.0}$ | – |
| Schedule-free | $93.6^{\pm 0.2}$ | $93.7^{\pm 0.1}$ | $71.1^{\pm 1.0}$ | $60.2^{\pm 0.8}$ | $56.3^{\pm 0.2}$ |
| LODE (ours) | $\mathbf{93.8}^{\pm 0.2}$ | $\mathbf{93.9}^{\pm 0.3}$ | $\mathbf{74.9}^{\pm 0.9}$ | $\mathbf{74.5}^{\pm 0.2}$ | $\mathbf{59.8}^{\pm 0.2}$ |

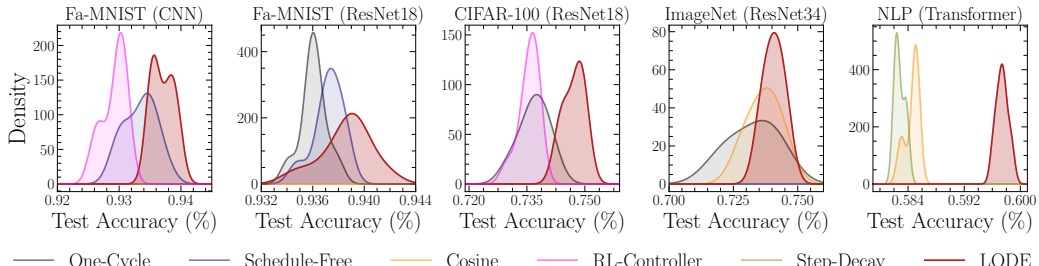

Figure 3: Probability distribution of the final test accuracy for the test problems from Table 1. We run 20 trials for each schedule, varying the random seed for the network initializations, after choosing the best performing hyperparameters for each schedule. The test accuracy is measured at the epoch with the highest validation accuracy. Colors indicate the scheduling or optimization method; for clarity, we only show the three best performing methods in each panel.

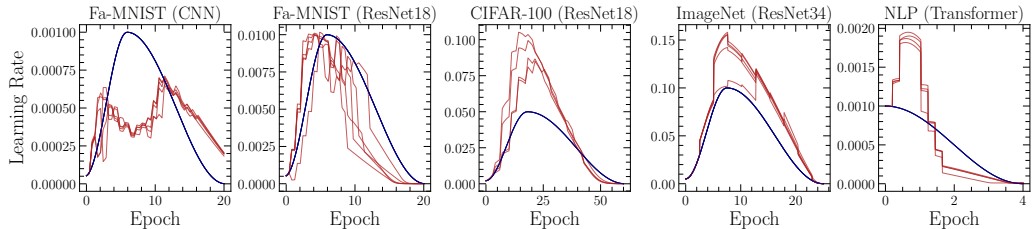

Figure 4: Learning rate schedules generated by the LODE scheduler (red) for 5 random trials on the test problems from Table 1, compared to the best parametric schedule for each task (blue)

Table 2: Largest eigenvalue of the Hessian of the loss function for ResNet-18 on CIFAR-100, computed via the power-iteration method. All values calculated for the best-performing models.

| Schedule | LODE | OneCycle | Hypergrad | Schedule-free | Cosine | Decay | Const |
|---|---|---|---|---|---|---|---|
| $\lambda_{\max}$ | $0.62^{\pm 0.22}$ | $1.22^{\pm 0.44}$ | $2.36^{\pm 0.37}$ | $2.66^{\pm 0.29}$ | $3.00^{\pm 0.29}$ | $3.69^{\pm 1.53}$ | $13.23^{\pm 3.56}$ |

We compute the sharpness of the minima achieved by each model by calculating the largest eigenvalue ($\lambda_{\max}$) of the approximate Hessian of the loss function with respect to the model parameters after optimization, using constant batch sizes of 256. The results are listed in Table 2. The LODE scheduler produces $\lambda_{\max}$ at least factor two smaller than other schedulers evaluated at their best performance. The LODE scheduler therefore not only achieves superior final validation performance but should also improve model generalization.

## 5.4 ABLATION STUDIES

**Configuration parameters** As described in Section 3.2, the LODE scheduler contains four configuration parameters: $n, \sigma, \mu, \Delta t$, representing the ensemble size, ensemble noise level in latent space, encoding length, and forward-looking horizon, respectively. We show the sensitivity of our scheduler to changes of the first two parameters with an ablation study performed with the CIFAR-100 dataset and the ResNet18 model in Table 3. We see that ensemble size and scatter have no significant effect on the scheduler performance and set defaults to $n = 30, \sigma = 0.15$. The LODE performance test (Section A.3) indicates that $\mu$ as low as 5% of the optimization epochs suffices for a high-fidelity reconstruction of the training dynamics, which we adopt as the default.

Table 3: Validation accuracy averages over 5 runs on CIFAR-100 when varying LODE configuration parameters.

| $n$ | $\sigma$ | Val Acc (%) $\uparrow$ |
|----|------|-----------------------|
| 10 | 0.15 | $74.2^{\pm 1.2}$ |
| 10 | 0.30 | $73.7^{\pm 1.1}$ |
| 30 | 0.15 | $74.9^{\pm 0.9}$ |
| 30 | 0.30 | $73.6^{\pm 1.4}$ |
| 60 | 0.15 | $74.8^{\pm 1.2}$ |
| 60 | 0.30 | $73.9^{\pm 0.7}$ |

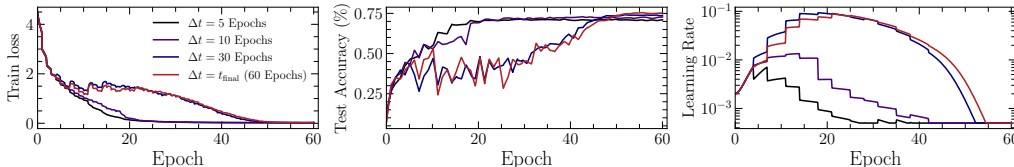

Figure 5: Training, test performance, and learning rate of the LODE-scheduler when changing the forward-looking horizon $\Delta t$ in the ResNet18–CIFAR-100 trial.

**Horizon** With these defaults, we test the most important parameter, $\Delta t$, which determines how far in the future the estimated validation performance is evaluated (step 4 in Algorithm 1). We vary the range from the minimum $\Delta t = \mu$, seeking to achieve the best performance at the end of the next training segment, to the maximum, $\Delta t = T - t_{\text{current}}$, delaying optimal performance to the very end of the optimization. The results are shown in Figure 5, again for training the ResNet18 model on CIFAR-100. It is evident that greedy adjustments are inferior to adopting a long-term goal. A long horizon permits a longer period of exploration with drastically higher learning rates, which are reduced only shortly before the final epoch $T$ to permit a more gradual convergence. This long-term goal orientation is crucial for the superior performance of our scheduler and would be impossible to achieve without an accurate long-term prediction of the training dynamics by the LODE.

**Latent ODE training data** We also test the performance of the LODE scheduler as a function of the quantity and quality of the training data obtained during the hyperparameter search. This study is important to determine the upfront cost of training the LODE model or the possibility of simply exploiting existing experiment tracking results without any further generation of training data. We therefore assume a simple grid search with 4 parametric schedule shapes (constant, OneCycle, cosine decay, exponential decay) and 5 overall learning rates. Table 4 lists the most important results. Increasing the number of random network initializations with fixed hyperparameters mildly increases the LODE model's accuracy and thus the scheduler's efficacy, presumably because it illustrates the stochastic effects on the training. The largest impact stems from the presence of the OneCycle schedule in the training data: removing it leads to a significant drop in performance. But only using OneCycle during LODE training reproduces exactly the performance of the OneCycle schedule from Table 1. We conclude that a modest amount of experiments during the hyperparameter search will suffice to create a useful representation of the training process. All results in this work are therefore based on 100 training runs (5 random seeds, 4 schedules, 5 overall learning rates).

Table 4: Test accuracy in CIFAR-100 when varying data volume (the number of random initializations of the test network at fixed hyperparameters) and schedule diversity to train the LODE model.

| | 1 init | 5 inits | 10 inits | no OneCycle | only OneCycle |
|---|--------|---------|----------|-------------|---------------|
| Test Acc (%) $\uparrow$ | $73.9^{\pm 0.9}$ | $74.9^{\pm 0.9}$ | $74.8^{\pm 0.9}$ | $72.4^{\pm 1.5}$ | $74.0^{\pm 1.1}$ |

## 5.5 COMPUTATIONAL COST

The LODE scheduler incurs two additional costs compared to simple parametric schedules: the initial training of the LODE model, and, at test time, the integration of training dynamics for each member of the latent ensemble. The former is negligible given the cost of the hyperparameter search for any moderately large training model. As demonstrated in Section A.3 no hyperparameter tuning for the LODE model is needed to achieve accurate reconstructions with the total walltime using a single NVIDIA A100 coming in at well under one hour. For comparison, the total training time for the RL-controller proposed by Xiong et al. (2022) is roughly twice as long when tested on the ResNet18 on the Cifar-100 task. Table 5 list the walltime per epoch at test time for the LODE-scheduler with a range of hyperparameter configurations compared to the other schedules and optimization methods, where $\mu$ indicates how many LODE updates are performed per training epoch. With default parameters, the LODE schedule is about $25\%$ more expensive than simple parametric schedules, but significantly cheaper than hypergrad optimization and the RL scheduler. It is also important to note the inference cost of the LODE does not scale with model size (unlike RL methods), hence for larger more costly models the extra cost of running the LODE-scheduler will become insignificant.

Table 5: Walltime per epoch for different schedules and optimization techniques in the ResNet18–CIFAR-100 trial on a single NVIDIA-A100. Runtimes are averaged over 5 trials of 20 epochs. The update frequency $\mu$ is reported in percentage of total epochs, $n$ denotes the latent ensemble size.
[*] There is no current official public implementation of the RL-schedule method from Xiong et al. (2022), we report numbers from our implementation of the partially available code.

| | LODE Scheduler | | | | | | Parametric | Hypergrad | Schedule-Free | RL-Schedule |
|---|---|---|---|---|---|---|---|---|---|---|
| $\mu$ | 10% | 5% | 2.5% | 10% | 5% | 2.5% | – | – | – | – |
| $n$ | 20 | 20 | 20 | 40 | 40 | 40 | – | – | – | – |
| Time (s) | $5.11^{\pm .1}$ | $5.47^{\pm .1}$ | $6.09^{\pm .1}$ | $5.98^{\pm .1}$ | $6.12^{\pm .1}$ | $6.28^{\pm .1}$ | $4.79^{\pm .0}$ | $11.68^{\pm .2}$ | $5.01^{\pm .1}$ | $11.2^{\pm .4}$ |

## 6 DISCUSSION

By learning from the training behavior a test network exhibits during a conventional hyperparameter grid search, we can 1) determine an effective representation of the current training state, 2) predict with high accuracy the future performance when following a proposed schedule, and 3) create a specialized hybrid schedule to achieve the best predicted long-term validation performance under a given optimization objective. Following this schedule indeed leads to minimizers with superior validation results, located in smooth regions of the loss function.

This outcome is surprising for two reasons. Any dynamical model requires a suitable state representation, but our approach constructs this state representation from three simple quantities that are routinely tracked in ML experiments. Unlike the RL-scheduler of Xu et al. (2019); Xiong et al. (2022), information about the parameters of network is not needed. The LODE state therefore represents the *effective* training behavior, which is legitimate because in deep learning architectures all minima are practically equivalent (Choromanska et al., 2015; Ge et al., 2016). Our approach can be applied to test networks of arbitrary size and is less computationally expensive and more data efficient because it does not seek to learn the dynamics of the network parameters, only of the aggregate network performance. The latent representations thus indicate training *phases*: e.g. the initial phase of gathering a reliable gradient direction; an exploration phase with high volatility; a convergence phase where the loss becomes shallow; and possibly an overfitting phase with divergence of training and validation performance (see Figure A1 for a visual confirmation of these training phases). Recognizing these phases allows the scheduler to adapt the learning rate such that it achieves high validation performance at the end of the optimization period.

The second surprise lies in the stability of the optimization despite the large learning rates that are, at certain times, proposed by the LODE scheduler. This empirical finding is consistent with works by Cohen et al. (2021); Arora et al. (2022), who also find that learning rates for deep neural networks operate at and often substantially exceed the limits of traditional stability: For quadratic loss functions, gradient updates remain stable only for learning rates $\eta < 2/\lambda_{\max}$, where $\lambda_{\max}$ is the largest eigenvalue of the Hessian of the loss function. Figure 6 shows the relevant quantities for all schedules in the ResNet18–CIFAR-100 trial. The horizontal line in the third panel indicates the traditional "edge of stability" (EoS) criterion. Several schedules track this line initially, but the

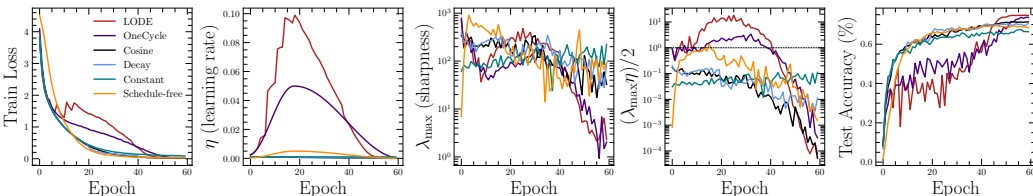

Figure 6: Training loss (1st panel), Learning rate (2nd panel), loss sharpness (3rd panel), optimization stability (4th panel), and test accuracy (5th panel) for different schedules in the ResNet18–CIFAR-100 trial.

LODE scheduler and, to a lesser degree, the OneCycle schedule exceed the EoS limit in the middle phase of the optimization. Not only do both remain stable, their investment in a longer exploration phase yields better long-terms results than those schedules with smaller $\eta$.

The second panel of Figure 6 shows the time evolution of the sharpness of the loss, revealing a noticeable reduction after $t \approx 40$, indicating increasing model generalization at same the time when the learning rate is decreased and the validation accuracy makes the most important late-time gains. While earlier works (Jastrzkebski et al. 2017, 2018; 2020) have demonstrated that large learning rates lead SGD into areas of the loss function with lower sharpness, the picture that now emerges for network training is more nuanced. Because loss landscapes for neural networks exhibit multi-scale properties (Ma et al., 2022), it is favorable to traverse the loss landscape with a sufficiently high learning rate to find the largest, flattest basin nearby. Not just is the interior of such a basin less rough, the number of wide and well-connected minima it contains scales with a power-law of its size (Ly & Gong, 2025). But traversing the loss landscape at that speed leads to highly non-monotonic losses, for which only a long-term average is guaranteed to decrease (Arora et al., 2022). To ensure good final performance for a finite-length optimization, learning rates must decline to reduce the stochastic component of the loss. The LODE scheduler recognizes these different phases and predicts when and how the transition from exploration to convergence phase needs to occur.

We surmise that an optimizer with our schedules is better able to find and settle in large basins, which explains the significantly reduced sharpness in Table 2. This ability appears to become more important in higher dimensions, where the basins occupy a smaller fraction of the volume, as shown from the largest improvements in Figure 3 being attained for the largest model, the transformer.

## 7 CONCLUSION

We present a generative learning rate scheduler, which uses metrics commonly gathered during hyperparameter searches to create a dynamical systems representation of the neural network training process. By leveraging the entire time evolution of the optimization under different schedules— instead of merely the final outcome—our method assembles an optimal schedule for a given model– task–dataset combination. It predicts which schedule will lead to the best long-term performance and achieves SOTA test accuracy and model generalization across image classification and text generation tasks. The gains from employing this scheduler become more pronounced for larger networks and more complex loss function. Our method is computationally efficient, numerically stable, and creates new, specialized paths to superior results in high-dimensional gradient descent.

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

# A APPENDIX - EXPERIMENTAL DETAILS

## A.1 MODEL ARCHITECTURES

**CNN** The CNN is designed for Fashion-MNIST dataset, we apply two consecutive $3 \times 3$ convolution–ReLU layers (64 channels) followed by max pooling and dropout ($p = 0.25$), then repeat the pattern with 128-channel convolutions. The flattened features pass through a 256-unit fully connected layer with ReLU and dropout before a final dense layer.

**ResNet** We follow the implementation from (He et al., 2016) creating an 18-layer ResNet for use on the Fashion-MNIST and CIFAR-100 datasets, and a 34-layer ResNet for the ImageNet dataset.

**Transformer (miniGPT)** We train a miniGPT model (Radford & Narasimhan, 2018) for our NLP task. Each Transformer block applies masked multi-head self-attention followed by a feed-forward MLP, with dropout ($p = 0.1$) and residual connections around each sub-layer. Layer normalization is applied after each residual sum. We use 8 transformer blocks with 8-head attention and 256-dimensional embeddings over sequences of length up to 256.

## A.2 BASELINE DETAILS

We compare against parametric, hypergradient, schedule-free and an RL baseline. We perform hyperparameter sweeps for each baseline with all results presented in the main paper coming from the best performing models.

**Parametric models** We use the built in learning rate schedules from OPTAX for cosine-decay, cosine-OneCycle, step-decay and constant learning rate schedules. For the Fa-MNIST and CIFAR-100 trials we perform sweeps over the learning rate (initial for cosine-decay and step-decay, peak-lr for OneCycle) for $\eta \in \{1\mathrm{e}{-3},\ 5\mathrm{e}{-3},\ 1\mathrm{e}{-2},\ 5\mathrm{e}{-2},\ 1\mathrm{e}{-1}\}$. We use ADAMW for all trials and

try both with, and without Nesterov momemtum finding no significant differences. For the CNN models on the Fa-MNIST data we use weight decay of $1e{-}3$, for the ResNet18 on Fa-MNIST we use weight decay of $1e{-}4$, and for the ResNet18 on CIFAR-100 we use weight decay of $1e{-}2$ for all trials.

For the ResNet34 on the ImageNet dataset we use standard SGD with Nesterov momemtum in all trials and a weight decay of $1e{-}4$. We sweep over higher learning rates than the other experiments with $\eta \in \{0.005,\ 0.01,\ 0.05,\ 0.1,\ 0.256\}$.

Once we find the optimal hyperparameters for each parametric model we perform 10 additional trials varying the random seed for both weight initializations, and training data shuffling.

**Hypergradient descent** We implement hypergradient descent as in (Baydin et al., 2017) to calculate the individual learning rate updates. We perform the same $\eta$ sampling as in the above parametric models for the Fa-MNIST, CIFAR-100 and ImageNet datasets where $\eta$ represents the initial flat value of the learning rate used.

**Schedule-free** We use the schedule-free learning rate implementation directly from the OPTAX library, ensuring to set the underlying $\beta_1$ values in the adamW optimizer initially to 0, resetting them to the optimal value we found 0.9 in the instantiation of the schedule-free optimizer. We perform the same $\eta$ sampling as the above for all datasets. We make sure to use the correct evalutation parameters as explained in the documentation via running `eval_params = optax.contrib.schedule_free_eval_params(opt_state, params)` before assessing validation, and test performance.

**RL-controller** We implement the RL-controller explained in (Xu et al., 2019) and detailed here URL. We made minor changes to resolve version conflicts with outdated python packages. We again perform the same hyperparameter sweeps as with the other baselines. We note that, due to extremely long training times, we do not train this model on the ResNet34, ImageNet, or transformer model.

A.3 LODE ARCHITECTURE AND TRAINING

We train a latent ODE (LODE) model for each dataset–model combination, resulting in 5 final trained LODEs. The training data for each LODE consists of the training trajectories of the parametric schedulers. We use 5 random trails for each schedule, resulting in 100 training trajectories for each LODE model. For simplicity, and to demonstrate efficient use of this method without any hyperparameter tuning of the LODE model itself, we use identical architectures for all trained LODEs with a single training run per model. We use latent and hidden dimensions of 20, and the ODE function is modeled with an MLP with 2 layers of 20 units with Tanh activations. We use an ODE-RNN encoder as in Rubanova et al. (2019) however we remove the variational penalty as in (Sampson & Melchior, 2025). We train with a batch size of 20 and use a OneCycle schedule with a peak learning rate of 0.001 using ADAM. The maximum training time of the LODE models used 50,000 gradient updates, taking just under 1 hour on a single NVIDIA A100 GPU, all 5 models converged below 1 hour of GPU time, and we performed nu hyperparameter tuning.

To determine the efficacy of our approach it is important to validate the reconstruction fidelity of the trained LODE model. The reconstruction accuracy of the trained LODE models for each trial is shown in Table A1. We see very accurate predictions even when using as little as 5% of the data to extrapolate the full training trajectories.

**Ability to select optimal trials** We also determine if our model is able to identify the optimal performing learning rate schedules from the data. To test this we supply the latent ODE with the first epoch of training data for each parametric schedule configuration and then ask the model to extrapolate, which trial will have the largest validation accuracy at the final epoch. We perform this test for each dataset–model combination, listing the ranking of the final validation accuracy of the chosen trial (out of 100 trials) with the final validation accuracy of that trial, as well as the final accuracy of the single best trial. As we are not averaging over seeds, the maximum final accuracy reported here are slightly higher than in Table 1. In all tests, we correctly ranked the best performing hyperparameters, the results below show our ability to adequately distinguish between best performing individual seeds withing each hyperparameter trial.

Table A1: Relative MSE for latent ODE predictions averaged across 5 random seeds per schedule for all dataset–model tests when observing either the initial $5\%$ or the initial $50\%$ of the training time series.

| Metric | Fa-MNIST (CNN) | Fa-MNSIT (ResNet18) | CIFAR-100 (ResNet34) | ImageNet (ResNet34) | NLP (Transformer) |
|---|---|---|---|---|---|
| train loss (5%) | $0.177^{\pm.041}$ | $0.105^{\pm.025}$ | $0.028^{\pm.006}$ | $0.021^{\pm.003}$ | $0.036^{\pm.011}$ |
| train loss (50%) | $0.119^{\pm.028}$ | $0.106^{\pm.025}$ | $0.024^{\pm.006}$ | $0.014^{\pm.002}$ | $0.008^{\pm.002}$ |
| learning rate (5%) | $0.287^{\pm.074}$ | $0.340^{\pm.081}$ | $0.335^{\pm.078}$ | $0.349^{\pm.051}$ | $0.231^{\pm.052}$ |
| learning rate (50%) | $0.241^{\pm.057}$ | $0.151^{\pm.081}$ | $0.132^{\pm.078}$ | $0.100^{\pm.011}$ | $0.131^{\pm.050}$ |
| val accuracy (5%) | $0.041^{\pm.007}$ | $0.031^{\pm.001}$ | $0.011^{\pm.003}$ | $0.020^{\pm.003}$ | $0.051^{\pm.013}$ |
| val accuracy (50%) | $0.010^{\pm.002}$ | $0.003^{\pm.001}$ | $0.011^{\pm.003}$ | $0.016^{\pm.002}$ | $0.021^{\pm.003}$ |

- FaMNIST (CNN): selected $2^{nd}$ best trial; final accuracy $94.3\%$; best final accuracy $94.3\%$
- FaMNIST (ResNet): selected $10^{th}$ best trial; final accuracy $93.9\%$; best final accuracy $94.3\%$
- CIFAR100: selected $4^{th}$ best trial; final accuracy $73.8\%$; best final accuracy $74.1\%$
- ImageNet: selected $2^{nd}$ best trial; final accuracy $74.4\%$; best final accuracy $74.7\%$
- Transformer: selected $2^{nd}$ best trial; final accuracy $57.46\%$; best final accuracy $57.47\%$

We see that via observing the training metrics of first epoch, our trained LODE models perform very well in determining which trial leads to the best final validation accuracy. This result, combined with the accurate reconstruction of the learning rate schedules demonstrated in Figure 2 and Table A1, give us confidence that the trained LODE models are learning useful latent representations that capture the underlying training dynamics and can reliably guide learning rate selection from early signals.

## A.4 CHOICE OF EVALUATION METRICS

To determine the *optimal* trajectory we choose validation accuracy rather than validation loss for two reasons. 1) In general, the validation accuracy provides a smoother curve, making LODE training and inference easier. 2) In most cases one is more interested in the final accuracy (or accuracy-related metrics such as the F1 score) than the loss. As the gradient updates are already guided entirely by minimizing the loss, including information about validation accuracy in the optimization routine is preferable.

## A.5 LATENT REPRESENTATIONS

In Figure A1 show a UMAP (McInnes et al., 2018) of the latent encodings created by our LODE encoder for 20 random trials of training a ResNet18 on CIFAR-100. We show results for 3 parametric schedules of constant, cosine-decay and onecycle, as well at the embedding from trials using the LODE, and show 10 points per epoch per trial. We note stark differences between the latent embeddings for the constant, and cosine schedules compared to the onecycle and LODE schedule. We can also see that the LODE model finds distinct clusters corresponding to the initial optimization phase, the exploration phase, and the convergence phase. These representations allow the LODE scheduler to shape the optimization in such a way to achieve the best long-term performance.

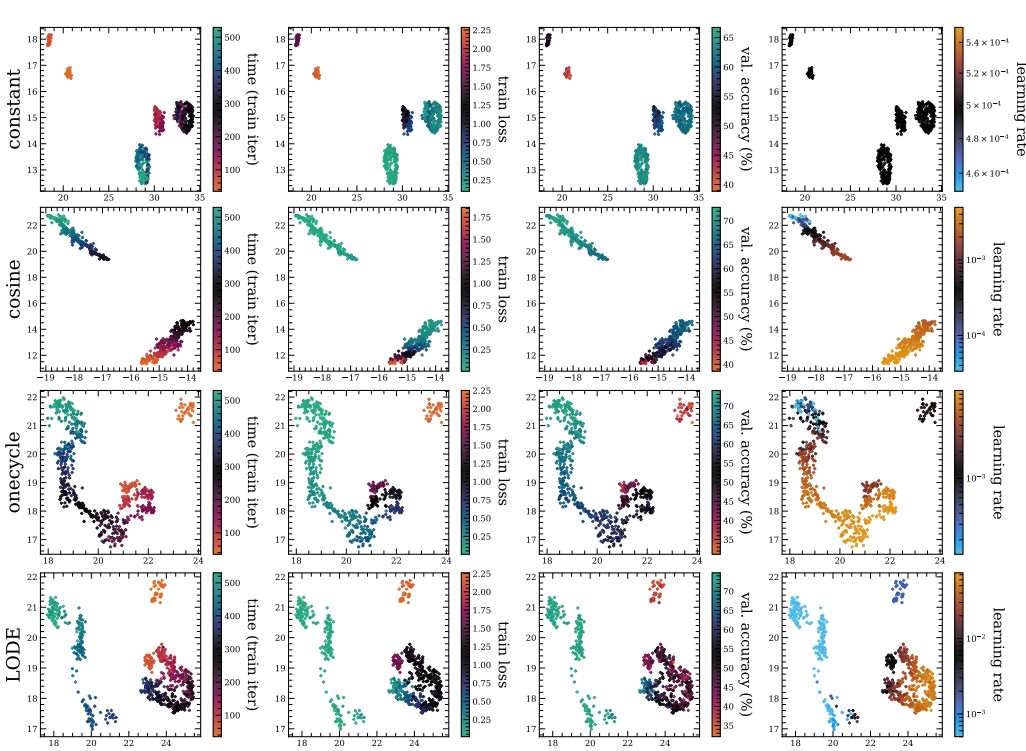

Figure A1: UMAP of the latent vectors generated by the LODE-encoder for a series of parametric, and LODE-guided trials of training on the ResNet18–CIFAR-100 trial. We show the training time, train loss, validation accuracy, and learning rate.

