# OpenReview forum: "Dynamics of Learning: Generative Schedules from Latent ODEs"
_ICLR.cc/2026/Conference — Submitted to ICLR 2026_

### Official Review · Reviewer_DnwH · 2025-10-28

**Soundness:** 2
**Presentation:** 2
**Contribution:** 2
**Rating:** 4
**Confidence:** 3

**Summary:**

*Disclaimer *  I used chatgpt for better **understanding**  details of sec 3.2.  and nothing else.

The authors propose a meta learning approach for learning learning rate/step size for first order optimization methods ( i.e. SGD and variants) typically deployment in deep learning.  The idea is to use a pretrained latent ode model that capture training dynamics and model the future trajectories include the triplet (training loss, validation metric, learning rate) from current time * to the end T.  The latent ode model is pretrained using a variety of schedulers. The proposed approach demonstrate the superior test acc on 4 benchmarks in vision + NLP.  It also shows the computation cost in wall clock is lower compared with other schedulers.

**Strengths:**

A few strengths include 1). The thoroughness of experiment to support major claim including the experiment accuracy and wall clock efficiency compared to baselines. 2). The topic is interesting/relevant to ml optimization especially in large scale dl systems.

**Weaknesses:**

A few weaknesses include

1): approach is not novel – NODE has been use before for similar tasks such as area of learning curve exploration( e.g. Architecture-Aware Learning Curve Extrapolation via Graph Ordinary Differential Equation , ding et al. aaai 25) .  I think there are other relevant papers such as Hop, skip, jump to Convergence: Dynamics of Learning Rate Transitions for Improved Training of Large Language Models emnlp 24

2) approach is less grounded since training loss curve is general not smooth, and neural ODEs require some Lipschitz smooth.

3).  The last point can be either pro or con, pro is simplicity of approach that is using the only triplet but not including any weight or graph structure of neural network.  the con is loss of potential information for modeling / future trajectory.

**Questions:**

1.	Can the author provide some insight to address weakness 2 training loss curve is general not smooth, and neural ODEs require some Lipschitz smoothness for well postedness and stability? However the loss curves are generally not smooth, so the metric value for validation, and learning rate can also be not lip-smooth. How can you justify the well-posedness and stability for your neural odes.


2.	Efficiency. I think the analysis of efficiency is to support one of your major claims.  I think in the proposed approach which requires some training using transformer model for your encoder and decoder which should account for/toward total wall clock times but it is not.  The authors mention it “is negligible given the cost of the hyperparameter search for any moderately large training model. “ If i understand correctly other baselines compared in the table are using some hyperparameter search.  nontheless, the pretraining should be included. Also some non wallclock or quantative analysis will support better for your claim.

3.	Accuracy. I also have a question about generalization. Since LODE is pretrained from all scheduler. Now in terms of test model generalization/accuracy, does it just memorize pretrain trajectory with certain schedulers  that best performing in terms of validation loss ?  The proposed approach may be more powerful if it is online. Some strategy like Learning to Boost Training by Periodic Nowcasting Near Future Weights paper.

---

> ### Author Response · Authors · 2025-11-15
>
> ## Weaknesses
> 1. Our method is novel, and the papers cited by the reviewer support this claim. Learning Curve Extrapolation (as in Ding+2025) shows that the final performance can be predicted from its early trajectory, but only
> uses this capability to rank the best tested hyperparameters. Our method generates unseen learning rates,
> uses the LODE to predict their performance (like Ding+2025, whose approach we replicate in Appendix
> A.3), and continually updates its prediction in response to the actual training results. Ding+2025 showed
> that a prediction of final outcome can be done, we’re using this insight to actually perform the learning
> rate adjustment dynamically.
>
> Hop-skip-jump (EMNLP24) analyzes the effect of a predefined change to the learning rate schedule and
> studies how this affect training dynamics. That paper concludes by saying ”*Our analysis provides insight
> into how learning rate schedules affect the training dynamics, and could inform the design of new schedules
> to accelerate convergence.*” Our work takes precisely this next step.
> We thank the reviewer for pointing out these two works and will include those citations in the revised
> manuscript to provide context for our contribution.
>
> 2. We disagree. While the neural ODE function cannot generate non-smooth dynamics, the state representation can be stochastic/noisy. The initial NODE paper (Chen+2018) successfully dealt with noisy input
> data to produce smooth reconstructions (see, e.g., their Fig 8). In addition, the transfer to a latent space
> can provide further robustness to noise through the encoder. Ding+2025 also demonstrated the use of a
> latent ODE model (through a graph state encoder) for stochastic losses.
>
> 3. In our opinion, simplicity is never a weakness. Our LODE training is lightweight, eminently feasible,
> independent of the size of the target model, and demonstrably effective.
>
> ## Questions
> 1. As we stated above, NODE and LODE models can be applied to non-smooth data. Their stability can be
> improved by suitable encoder/decoder architectures and the avoidance of the customary variational loss
> (e.g. Sampson+2024, arXiv:2410.08923). For a concrete example, see our Fig. A1 in the appendix.
>
> 2. We use a lightweight RNN as the encoder for the LODE network, not a transformer (see sentence after
> Eq. 2). As a result, the LODE training is several orders of magnitude faster (single GPU walltime easily
> below 1h) than the creation of the training data from optimization runs of the target model.
>
> 3. Our approach generalizes: the learning rates schedules it generates can deviate significantly from the
> best training example (see our Fig. 3). Similar to the cited paper by Jang+2023, our method is online.
> Instead of predicting the outcome of multiple gradient steps during the optimization, we predict the
> future loss of adjusting the learning rate. We will add a citation to the Jang+2023 paper to provide
> context for our work.

---

### Official Review · Reviewer_VrmJ · 2025-10-31

**Soundness:** 2
**Presentation:** 2
**Contribution:** 2
**Rating:** 2
**Confidence:** 3

**Summary:**

The paper considers modeling the learning rate schedule from trajectories of training metrics (loss, validation accuracy, etc.) of prior runs. The approach is evaluated on image classification and next token prediction settings and compared to standard hand-crafted learning rate schedules.

**Strengths:**

The paper studies an interesting problem that has been largely overlooked by the community. The initial experiments are promising.

**Weaknesses:**

- Experimental evaluation is overall limited. The authors include multiple datasets (FashionMNIST, CIFAR100, ImageNet) and models (ResNet-18/34) which is great. However, many of these datasets and models are fairly simple and make it hard to distinguish between different approaches (e.g., all of the tested methods perform similarly on FashionMNIST). It would be good to include evaluations with more challenging models and datasets. For example, in the case of image classification, showing results with ViT and ResNet-50 models on ImageNet would make it easier to compare against existing baselines.
- It would be good to include simple baselines that are based on the collected dataset of training runs. The paper currently compares hand-crafted schedules like constant or cosine to the proposed approach. It would be good to include a few simple baselines that e.g. learn linear weights between existing schedules. This would help understand how much of the gains are due to using prior data vs the specific method for using prior data.
- If I understand correctly from the last sentence of Section 5.3, the dataset used to model the learning rate schedule consists of 100 training runs. That sounds fairly small and it would be good to perform careful experiments to understand the level of generalization with respect to the training data. These could include nearest neighbor retrieval baselines or similar strategies to understand the level of memorization, interpolation, and generalization.
- It would be good to include much more details on the training data composition in terms of (task, model, schedule, performance). Related to my previous comment, it is a bit hard to interpret the current results without that.

**Questions:**

Please see the weaknesses section.

---

> ### Author Response · Authors · 2025-11-15
>
> ## Weaknesses/Questions
> 1. FashionMNIST, CIFAR-100 and ImageNet are standard benchmarks, which many papers have used and
> reported results for. ResNet34 has ∼ 21.7 million parameters, so it is unclear why ResNet50 with ∼ 25.5
> million parameter would be much different. We also test a transformer for text generation to demonstrate
> the success of our method on a different task and data modality. As shown in Table 1 and Figure 2, our
> results consistently outperform alternative schedulers in every test case.
>
> 2. Cosine and constant schedules are not hand-crafted, they are commonly employed, and in the case of NLP
> transformers even SOTA. We actually tested the idea proposed by the reviewer of an accuracy-weighed
> linear combination of the training schedules on the CIFAR-100 trial, and it achieves a test accuracy of
> 65.5 ± 1.7, whereas our method achieves 74.9 ± 0.9. The missing element of static schemes like this is
> that they do not respond to the learning phase the target model is currently in; our method does. As our
> Fig. 4 shows, exploiting the expected dynamics of the optimization is critical for good performance.
>
> 3. We found LODE training with our modified training objective very robust, even with a limited training
> data set. In fact, we want the training to be computationally cheap to make adoption of our method easy.
> The relevant test is whether we can predict the outcome of optimization, and Fig A1 demonstrates that
> we can.
>
> 4. All relevant details are in the appendix, and Table 1 shows full results for every task and model
> combination. If any further details would support our presentation, we’d be happy to provide them.

---

### Official Review · Reviewer_Lmnb · 2025-11-01

**Soundness:** 2
**Presentation:** 3
**Contribution:** 2
**Rating:** 4
**Confidence:** 4

**Summary:**

The paper proposes a Latent ODE (LODE)–based learning-rate scheduler that learns a continuous-time latent dynamics model from routine training traces—training loss, validation metric, and learning rate—collected during a conventional hyperparameter grid search. At test time, the scheduler encodes the most recent window of traces, samples a latent ensemble, integrates the ODE forward for each sample to predict future trajectories, and selects/averages the top-K learning-rate candidates that maximize long-horizon validation performance for the next
𝜇
μ steps (Algorithm 1). On CNNs/ResNets (Fa-MNIST, CIFAR-100, ImageNet) and a small Transformer (TinyStories), the method consistently outperforms parametric schedules, hypergradient descent, schedule-free, and an RL baseline on final accuracy, while being ~25% slower per epoch than simple parametrics but much cheaper than hypergradient and RL. Notably, authors claim LODE inference cost does not scale with model size, unlike RL methods.

**Strengths:**

Long-horizon objective: The scheduler explicitly optimizes for future validation performance, enabling exploratory high-LR phases followed by late-stage annealing; greedy short-horizon variants underperform.


Simple inputs, broad applicability: Uses only three traces routinely logged in experiments; no access to model parameters is required.

Consistent wins across tasks: LODE beats all baselines across CNN/ResNet/Transformer settings with tight accuracy variance.

Reasonable overhead: ~25% more wall-time than parametrics, much cheaper than hypergradient and RL; plus cost said not to scale with model size.

**Weaknesses:**

Heavy reliance on the quality/diversity of traces from the initial hyperparameter search. Ablations show outcomes depend on schedule diversity (notably presence of OneCycle) and benefit mildly from more random initializations; authors ultimately train each LODE from 100 runs (5 seeds × 4 schedules × 5 LRs). This creates a data-collection dependency that practitioners must meet.

RL baseline choice and recency. The primary implemented RL controller is Xu et al. (2019); the paper mentions Xiong et al. (2022) but (per the note) does not run an official or up-to-date RL implementation and even flags that their used code may not be runtime-optimal. It also skips RL on ImageNet and the Transformer due to cost, while reporting ImageNet RL numbers from Xiong et al. (2022). Stronger comparisons should at least include Xiong-style controllers run under the same setup.

Scaling claim vs. transfer. Authors assert LODE inference cost does not scale with model size (unlike RL). But they do not evaluate RL transfer patterns (e.g., training the RL policy on a small model/dataset and reusing it on larger ones), and they train a separate LODE per dataset–model (no cross-model transfer is demonstrated). Thus, the paper hasn’t shown that LODE transfers across model sizes/datasets, whereas RL methods in principle can be trained small and reused at inference. The scaling claim is supported by their runtime table and the decision not to run RL on larger tasks, but transferability remains untested for both, with LODE clearly not demonstrated.

External validity limits. Training data for LODE excludes hypergradient/RL/schedule-free runs “to stay within conven

**Questions:**

Trace quality & minimal budget. What’s the minimal number and diversity of runs (schedules × LRs × seeds) to reach “near-paper” performance, and how does this vary by dataset/model? Current results suggest sensitivity to OneCycle presence and use 100 runs per LODE. Could you provide a practitioner recipe for “small-budget” settings?

RL comparison. Why not include a modern RL scheduler (e.g., closer to Xiong 2022) trained/evaluated under the same hardware/runtime budget? Today’s RL controllers can often be trained on small proxies and reused—how would that alter your “RL doesn’t scale” claim?

Transfer & reuse. You train one LODE per dataset–model. Can a single LODE trained on a small proxy transfer to a larger model or a related dataset (possibly with lightweight adaptation)? If not, what prevents reuse—latent space mismatch, encoder overfitting to specific loss dynamics, or something else?

---

> ### Author Response · Authors · 2025-11-15
>
> ## Weaknesses
> Weaknesses 1-3 are directly related to Questions 1-3, so we’ll respond to them below. For Weakness 4, the text
> has been cut off, but we believe the reviewer intended to ask why we didn’t use advanced methods such as
> RL or hypergrad during the LODE training. The answer is simplicity. We very intentionally want to keep the
> complexity and walltime of the LODE training phase low and potentially even use existing HP sweeps, which
> would unlikely have used such options. However, we test against each of these methods and show superior
> performance in every case in Table 1.
>
> ## Questions
> 1. We agree that training data for the LODE needs to be generated, but the demands can be met by many HP
> grid searches. In addition, our code contains a wrapper for the LODE training to be performed on-the-fly
> with the optimization of the target model. This allows us to implement (and possibly update) best use
> principles regarding schedule diversity and number of seeds. For instance, we have determined that a
> smaller number of overall learning rates and seeds is feasible, and (as we point out in the paper) that not
> all schedule shapes are necessary. So, cost reduction and ease of use of the LODE training are definitely
> attainable.
>
> 2. The code for the Xiong+2022 paper was removed from github, which is why we cited their reported
> test accuracy results (which our method comfortable beats). We recently found a zip file with a partial
> implementation attached to their paper’s OpenReview page and are in the process of getting it to run on
> our benchmarks to provide runtime results. We will post them here if we are successful (their code uses
> outdated packages and is customized specifically for RoBERTa training). In any case, our statement is not that ”RL doesn’t scale” but that our method’s training and runtime is by construction independent of
> model size and amounts only to a 25% runtime increase even for our smaller models.
>
> 3. While we have not demonstrated transfer of our schedules across different models or data sets in this
> work, we are not aware of any RL scheduler method having demonstrated that capability either. In fact,
> we believe that training small and testing large is likely to be successful for our LODE scheduler and
> therefore remains one of our objectives for the LODE training wrapper mentioned in Q1 above.

---

> ### Author Response · Authors · 2025-11-18
> **Xiong+22 benchmarking**
>
> We have implemented the RL controller proposed by Xiong+2022 and tested it on the ResNet18/CIFAR-100
> model/data combination. We find the graph-based RL controller takes over twice as long to train end-to-end
> as our method (using 30 episodes and updating every 500 steps as specified in their paper). More importantly,
> we find inferior results compared to our LODE scheduler (validation accuracy 73.9 ± 0.9, compared to 74.9 ±
> 0.9 from ours), confirming our previous statement based on their published results for the larger ImageNet
> test, which we also beat by a significant margin. We updated Table 1 and Figure 3 with these new results.
> Lastly, the runtime at inference is twice as high as for our LODE model. We updated Table 5 accordingly. While
> faster than the Xu+2019 method, the cost of summarizing the entire network as a graph substantially exceeds
> the cost of integrating a neural ODE in a low-dimensional latent space. In summary, we can now firmly state
> that the Xiong+2022 scheduler is neither faster to train, nor faster to run, nor capable of producing trained
> models that would rival ours.

---

### Author Response · Authors · 2025-11-18
**Paper and methods clarification**

The questions of the reviewers helped us to improve the clarity of the manuscript and better articulate the novelty of our contribution. We have updated the submission pdf, with additions in the section on related works and the main results table and the inference time table. We moved Figure A1 from the appendix to the main text (now Figure 2) to show the accurate performance of the LODE model to reconstruct/predict the training performance. This figure may have been missed by readers so far. We also add a new figure A1 to the appendix, which shows a UMAP projection of the latent representation generated by the LODE encoder, which supports our claim (in the discussion section) of distinct and recognizable training phases.

In response to the reviewers we summarize our findings as follows:

1. We present a novel generative latent ODE scheduler (LODE), which constructs an adaptive/online learning rate schedule based on the current training dynamics of a neural model on a given task.


2. The training data comes from a hyperparameter grid over different learning rates and parametric learning rate schedules. The main results use a total of 100 hyperparameter trials as training data. We consider this extra cost to be small given typical hyperparameter searches for production models. We are investigating if learning-rate scaling relations would allow us to train smaller models for the LODE model, and then scale up to larger models at test time. If successful, we will add this option to the LODE training wrapper function we provide with our code. In any case, alternative approaches, specifically the RL methods in Xu+19 and Xiong+22, also require existing training checkpoints to train their controller, which we find over twice as expensive to train and to run.


3. As far as we know, we are the first to present a generative approach to learning rate schedules. As pointed out by Ding+2025, a latent ODE method can predict late-time training performance and rank the chosen hyperparameters. We take the next step to leverage this insight and predict high-performing schedules.


4. We test our scheduler on a range of standard computer vision tasks (Fashion-MNIST, CIFAR-100, and ImageNet) with a variety of models (CNN, ResNet18, ResNet34). We also test our LODE scheduler on a transformer model for next-token prediction.


5. Our scheduler outperforms all alternative methods on every task.


6. The new learning rate schedules, as shown in Figure 4, are substantially different from any schedule seen during training.


7. Importantly, as the LODE scheduler requires only training metrics, the cost at training and inference is independent of model size. Even for smaller models, like ResNet18, the runtime overhead, compared to parametric schedules, is only about 25\%. For larger models, the extra inference cost of evaluating the LODE becomes negligible.


8. This work furthers our understanding of stochastic optimization. Figures 5, 6 and A1 demonstrate that distinct training phases exist, and that a pronounced exploration phase with learning rates far beyond the limits of traditional stability are not just possible, but highly beneficial for producing models with superior accuracy and generalization.

---

### Comment · Area_Chair_HM17 · 2025-11-25
**Please discuss further**

Reviewers, please read and respond to the author rebuttals. Have they addressed your concerns or clarified anything?

---

### Author Response · Authors · 2025-12-02
**Final message to new AC**

We would like to again raise concern about the quality of the initial reviews. The alleged weaknesses stem from a surface-level reading of the manuscript and have been fully addressed in our responses. Several claims of the reviewers are factually wrong, some lack any merit. Due to the disruption of the review process, we did not get the chance for a proper discussion, so we ask you to evaluate what we actually submitted: a new method to choose the learning rate that outperforms any existing schedule on every one of a range of standard image and language benchmarks.

---

### Meta-Review · Area_Chair_ojUt · 2026-01-06

**Summary:**

The paper proposes LODE, a latent-ODE–based scheduler that learns from prior training traces to choose future learning rates online. It encodes a short window of observed loss/validation/LR, simulates future trajectories in latent space, and selects the next LR by maximizing predicted long-horizon validation performance. Results on Fashion-MNIST, CIFAR-100, ImageNet (ResNet-18/34) and TinyStories show consistent gains over parametric schedules, hypergradient descent, “schedule-free,” and RL baselines, with roughly 25% training overhead vs simple schedules. The authors updated the submission with clearer positioning, stronger figures, and added an implementation of a more recent RL controller (Xiong et al., 2022) on CIFAR-100, reporting worse accuracy and higher cost than LODE.

Despite these positives, the evaluation remains limited. The strongest concerns are about external validity and baselines. The method relies on a pre-collection of 100 runs per dataset–model for training the scheduler; sensitivity to schedule diversity (e.g., presence of OneCycle) is acknowledged, but minimal budgets and robustness are not convincingly established. Comparisons to modern RL controllers are incomplete, especially on ImageNet and the Transformer; the updated CIFAR-100 comparison helps but does not resolve the broader claim. The paper makes a scaling argument that LODE inference cost is independent of model size, but no transfer across model sizes or datasets is demonstrated; each LODE is trained per dataset–model, and no proxy-to-large transfer is shown. Simple data-driven baselines that exploit the same traces (e.g., linear mixtures of schedules, nearest-neighbor retrieval of schedules) are only partially explored and do not fully isolate where gains come from. Finally, some related work on attention steering and context-faithful prompting is not directly relevant here; more pertinent is prior work on learning-curve extrapolation and schedule design; these connections are improved but still feel underdeveloped in terms of empirical differentiation.

**Reviewer Concerns:**

The authors addressed clarity issues, added figures, corrected some misreadings, and expanded related work. They moved an important reconstruction figure into the main text and added a UMAP analysis to support phase interpretation. They implemented Xiong et al. (2022) on CIFAR-100 and reported worse accuracy and higher cost, strengthening their case on that setting. They clarified that the encoder/decoder are RNN-based, not Transformers, and detailed runtime overhead. They also ran a linear-mixture baseline for CIFAR-100, which underperformed LODE.

Open issues remain. The reliance on 100 prior runs per dataset–model is substantial; the paper does not convincingly establish a minimal, practical budget across settings, nor does it quantify performance vs budget beyond a few qualitative statements. Transferability is not shown: training on a small proxy and reusing on larger models/datasets would substantiate the scalability claim and provide a fairer contrast with RL. The RL comparison is narrow: skipping RL on ImageNet and the Transformer while arguing scalability weakens the generality of the claim, even if cost is high. Stronger, matched-cost RL baselines or principled budget-normalized comparisons would help. Finally, broader model/dataset coverage (e.g., ViT/ResNet-50 on ImageNet) and more thorough data-driven baselines would improve external validity and interpretability.

**Reviewer Scores:**

Reviewers who were marginally positive on the promise and added experiments would likely remain at 4 given the persistent external validity and baseline gaps.

Reviewers focused on scope, transfer, and RL baselines would remain at 2–4. The CIFAR-100 RL update helps but is not sufficient to swing ImageNet/Transformer concerns.

Overall, I do not expect substantial upward movement without stronger transfer and matched-cost baselines.

---

### Decision · Program_Chairs · 2026-01-26

Reject